Urban areas promotes shifts in the proportion of prey consumed by four raptor species (Accipitridae) in Mexico

Márquez-Luna Ubaldo 1 marquezubaldo@iztacala.unam.mx
Rivera-Hernández Oscar 2
1 Facultad de Estudios Superiores Iztacala, Universidad Nacional Autónoma de México , Tlalnepantla de Baz, Estado de México , Mexico
2 Mexico City , Mexico
Sunny Armando
Electronic publication date: 2025 Oct 29
Publication date: 2025
Volume: 13
Electronic Location ID: e20307
Received 2025 Apr 15; Accepted 2025 Oct 8
Copyright: © 2025 Márquez-Luna and Rivera-Hernández
Copyright year: 2025
Copyright holder: Márquez-Luna and Rivera-Hernández
License: This is an open access article distributed under the terms of the Creative Commons Attribution License, which permits unrestricted use, distribution, reproduction and adaptation in any medium and for any purpose provided that it is properly attributed. For attribution, the original author(s), title, publication source (PeerJ) and either DOI or URL of the article must be cited.
License URL: https://creativecommons.org/licenses/by/4.0/

Keywords: Citizen science, Cooper’s Hawk, Common Black Hawk, Roadside Hawk, Gray Hawk, Urbanization, Web-sourced photography

Funding: The authors received no funding for this work.

==============================
Background

Urbanization is a process of environmental change which reduces and fragments the original habitat and creates new pressures, conditions, and resources for the species. The urban areas act as an ecological filter, which may exclude species, while others can colonize and thrive in cities, generating a process of biotic homogenization. As top predators, the changes in prey community composition could affect the diet of raptors. However, this effect has been understudied.

Methods

In this study we compiled information about the frequency of prey consumed by Cooper’s Hawk, Common Black Hawk, Roadside Hawk and Gray Hawk using two sources of information: (1) data extracted from photographs available on citizen science platforms and (2) a search of scientific literature. Using these data, we compared the diet composition, the proportion of prey consumed and the trophic niche breadth by four species of raptor in urban and non-urban areas. Additionally, we evaluated whether the variability in frequency of predation records by raptors was associated to the type of prey or their breeding season.

Results

Our results indicate that the raptors studied consumed at least 91 prey species in Mexico. Most records of predation by Common Black Hawk, Roadside Hawk, and Gray Hawk occurred in non-urban areas. On the contrary, most records of predation by Cooper’s hawks occurred within urban areas. There was no statistical difference in the richness of prey consumed by Cooper’s Hawk and Gray Hawk between urban and non-urban areas. In contrast, fewer prey types were consumed by Common Black Hawk and Roadside Hawk in urban areas than in non-urban areas. The proportion of prey types consumed in urban and non-urban areas was significantly different for Cooper’s Hawk, Roadside Hawk and Gray Hawk. Conversely, the proportion of prey types consumed by Common Black Hawk’s was similar in urban and non-urban areas. The four raptors studied increased the proportion of birds consumed in urban areas by between 2% and 36%, regardless of their preference for a specific type of prey. The trophic niche breadth of raptors was broader in urban areas than in non-urban areas, except for Cooper’s Hawk, which had a narrow trophic niche breadth in urban areas than in non-urban areas. The temporal variation in the frequency of predation records was influenced by differential factors such as the breeding season, the type of prey consumed, and the utilization patterns of citizen science platforms. Our results demonstrate that the four raptors studied increased the proportion of bird consumption in urban areas. This shift in raptors’ diets could be linked to the increased prevalence of diseases transmitted by feeding on infected birds in urban areas. This emphasizes the importance of conducting research to assess the prevalence and transmission rates of diseases that could threaten the conservation of birds and raptors in urban areas.

Introduction

Urbanization is an environmental change process linked to habitat loss and fragmentation, reduced connectivity between patches of native vegetation, and the introduction and proliferation of non-native species (Hansen et al., 2005). These environmental changes to the original habitat create specific conditions and resources within cities that can act as an ecological filter, excluding some species and favoring the presence of others, generating a process of biotic homogenization (McKinney, 2008; Silva, Sepúlveda & Barbosa, 2016). Urbanization can have adverse effects on raptors, including a decline in population densities, species richness, and eventually, local extinction (Kettel et al., 2018). In addition to habitat loss associated with urbanization, urban areas present threats to raptors such as collisions, electrocution, poisoning and disease (Solaro, 2018; Leveau et al., 2022). However, some raptor species can colonize, establish and thrive within urban areas; these raptors tend to be generalists with an opportunistic diet associated with the consumption of abundant prey in these areas (Boal & Dykstra, 2018). This could promote changes in the diet composition of raptors within urban areas; however, this has been poorly investigated in most raptor species (Ballejo & De Santis, 2013).

Diet analysis is one of the basic requirements for the understanding of the biology of species; it allows determining nutritional relationships, as well as their roles in the ecosystem (Royer et al., 2019; Zarei, Yazdi & Rajaee, 2021). Diet composition and prey consumption preferences of raptors can highlight their role as top predators and their effect on the population dynamics of their prey (Rectenwald et al., 2021), interspecific interactions with its competitors (Pande et al., 2018), intraspecific interactions between sexes (Panter & Amar, 2022), evidence of conservation risks (e.g., bioaccumulation of toxic substances or transmission of diseases; Ruiz-Suárez et al., 2014; Kunca, Smejkalová & Čepička, 2015), even this information is fundamental in rehabilitation and wildlife reintroduction processes (Arenas et al., 2020). Understanding the ecological functions of raptors and their interactions can help to better manage habitats and resources for their conservation (Mills, Soulé & Doak, 1993; Frixione & Rodríguez-Estrella, 2020). Historically, the study of raptor diet has been carried out by identifying the undigested remains of prey in pellets. However, this method tends to be limited to the breeding season and nesting sites where it is most feasible to collect pellets (Hiraldo et al., 1991). Recently, molecular techniques have been used that allow the accurate identification of raptor prey through the detection of DNA in fecess, pellets, or remains of prey from the surface of raptor beaks and talons (Bourbour et al., 2021; Hacker et al., 2021). However, both methods require the collection and processing of samples that can be difficult to obtain. Recently, the use of photographs available on citizen science platforms has been explored to determine the diet of raptors at broad spatial and temporal scales, complementing traditional approaches (Naude et al., 2019; Berryman & Kirwan, 2021; Panter & Amar, 2021, 2022; Kannan et al., 2023; Panter et al., 2024; Pantoja-Maggi, Medrano & Martínez, 2024).

The family Accipitridae is composed of 74 genera and between 250 and 255 species (Mindell, Fuchs & Johnson, 2018). Species within this family have a wide morphological variability that has allowed them to occur in a wide variety of habitats and exploit a wide range of resources (Winkler, Billerman & Lovette, 2020). The trophic niche of some accipitrids is generalized, feeding on a wide variety of prey (vertebrates and invertebrates); while other species specialize in the consumption of a specific type of prey (Winkler, Billerman & Lovette, 2020). In Mexico, there are 38 species belonging to Accipitridae (Chesser et al., 2024). However, most of the raptor species distributed in Mexico lack studies about their diet composition and their prey preferences. The research has been conducted in the north of the country analyzing the composition of pellets and has focused on the following raptor species: Osprey (Pandion haliaetus), Golden Eagle (Aquila chrysaetos), Bald Eagle (Haliaeetus leucocephalus), Burrowing Owl (Athene cunicularia), Peregrine Falcon (Falco peregrinus) and Aplomado Falcon (F. femoralis; Rodríguez-Estrella & Rivera-Rodríguez, 2006; Morales-Yañez, Rodríguez-Estrella & Gatica-Colima, 2023; Villalobos-Juárez et al., 2023). Additionally, there are isolated observations of different raptor species consuming prey previously unreported in the scientific literature (e.g., Rodríguez-Canseco et al., 2015; Nahuat-Cervera et al., 2020; Valencia-Herverth et al., 2022).

In this study, (1) we determined the diet composition of the following four species of accipitrids: Cooper’s Hawk (Astur cooperii), Common Black Hawk (Buteogallus anthracinus), Roadside Hawk (Rupornis magnirostris) and Gray Hawk (Buteo plagiatus). We selected these species due to their wide geographic distribution throughout Mexico and their contrasting natural histories. To achieve this, we use two sources of information: (1) data extracted from photographs available on citizen science platforms and (2) a search of scientific literature. (2) We determined the most preferred type of prey of each raptor species. (3) We compared whether the diet composition, the proportion of prey consumed and the trophic niche breadth of each species of raptor changed in urban and non-urban areas. (4) Finally, we evaluated whether the frequency of predation records by raptors changed throughout the year, with the type of prey or during breeding season.

In the context of biotic homogenization promoted by urbanization, we expect that (1) the prey type richness consumed by raptors will be lower in urban areas than in non-urban areas; (2) the proportion of prey consumed by raptors will be different in urban areas than in non-urban areas; (3) and the raptors trophic niche breadth will be more generalist in urban areas. Considering that the energy requirements of birds of prey are higher during the breeding season, and this may promote greater activity around the nest to provide food for the chicks (Eldegard et al., 2003; Panter & Amar, 2022), we expect the temporary increase in raptor predation records to be associated with their breeding season.

Materials AND methods

Study area

The United Mexican States (hereafter Mexico) is located between 11°58′07″–32°43′06″ North latitude and 94°26′02″–114°43′22″ West longitude. Mexico has an altitudinal range that reached 5,636 m asl; the country has a wide variety of climatic conditions that allow the establishment of a great biological diversity (Espinosa et al., 2008). These conditions have facilitated the establishment of the following habitats in Mexico (the percentage of each habitat’s coverage in Mexico is shown in parentheses): xerophytic scrub (37.62%), coniferous and oak forests (19.35%), deciduous tropical forest (13.77%), evergreen tropical forest (9.95%), grassland (8.17%), thorn forest (5.80%), subdeciduous tropical forest (3.24%), aquatic and subaquatic vegetation (1.18%), and tropical montane cloud forest (0.92%, Rzedowski, 2006).

Study species

Cooper’s Hawk: The Cooper’s Hawk plumage in adults is finely barred rufous ventrally and blue gray dorsally; has a pale nape and barred tail (black and blue gray bars) with a white tip (Rosenfield et al., 2024). The juvenile plumage is brown dorsally and white ventrally with thin dark streaks, in some cases a teardrop like pattern (Kaufman, 2005; Davis, 2014). The species distribution ranges from southern Canada to Central America (Rosenfield & Bielefeldt, 1993; Cartron et al., 2010). In Mexico, it is widely distributed from the northern border to the southeast, extending as far as Chiapas and Campeche (Rosenfield et al., 2024). Despite being closely associated with forest environments, Cooper’s Hawks exhibit marked plasticity in habitat use and often inhabit sparsely wooded areas including highly urbanized environments (Boal, 2018; Rosenfield et al., 2024). Their rounded wings and long tails enable maneuverability for ambush hunting, an adaptive trait for species that occur in dense forests. Their breeding season extends from late March to early July (González-Oreja, Zuberogoitia & Jiménez-Moreno, 2019; Rosenfield et al., 2024).

Common Black Hawk: The adult plumage is dark gray to black. The Common Black Hawks have black bills with pale yellow base and also have long yellow tarsi. Their tail is short, broad, squared and has one white bar. On the other hand, the juveniles plumage differs from adults, its dorsally brown and buff below, with white streaking on the neck, hind neck and back. Also, have narrow bands in the tail and patterned face (Wheeler & Economidy, 2003; Kaufman, 2005; Duffy, 2012). The species distribution ranges from southwest USA to northern South America (Jurado-Ruiz & Moreno-Contreras, 2020; Schnell, 2020). In Mexico, it is found along the Pacific coast, ranging from Sonora to Chiapas, and from Coahuila to the Yucatán Peninsula (Johnson & Schnell, 2024). The Common Black Hawk is a strictly riparian species, associated with marshes, mangroves, gallery forests, or streams up to altitudes above 1,800 m (Howell & Webb, 1995; Ridgely & Greenfield, 2001; Flesch, 2008). Their breeding season extends from late April to early August (Schnell, 1994; Jurado-Ruiz & Moreno-Contreras, 2020).

Gray Hawk: The adult plumage in both sexes is mostly pale gray with thin white bars below. Their tail is relatively long with black and white bars. A distinctive of the species is a brown iris and a yellow cere. Moreover, the juvenile plumage is brown barred and has a pale inner wing area (Stiles & Skutch, 1989; Vallely & Dyer, 2018; Astorga-Acuña & Mora, 2024). The species distribution ranges from southwest USA to northern Costa Rica (Stewart, Pyle & Clark, 2023). In Mexico, it is found along the Pacific coast, ranging from Sonora to Chiapas, and from Tamaulipas to the Yucatán Peninsula (Bibles, Glinski & Johnson, 2020). The Gray Hawk can be found in a wide variety of habitats, from scrubland to oak and pine forests. It can be found on the edge or interior of the forest (Stiles & Skutch, 1989; Rodríguez & Banda, 2015). Their breeding season extends from mid-April to mid-August (Bibles, Glinski & Johnson, 2020).

Roadside Hawk: The Roadside Hawk shows considerable geographic variation; to the date are 12 recognized subspecies. Most of the subspecies are similar, their adult plumage is gray to brown above, with a gray head, yellow cere, banded tail and tight rufous banding from the breast to the vent. The tail varies from gray through brown to rufous with four even blackish bands and a white tip (Ferguson-Lees & Christie, 2005; Barros et al., 2010; Garay & Marín, 2019; Bierregaard, Boesman & Kirwan, 2024). The juvenile plumage is broadly similar to the adult but browner and edged buff dorsally; tail with five or six narrower dark bands; head somewhat streaked; throat and chest more or less streaked whitish, rufous, brown or dusky depending of the subspecies (Ferguson-Lees & Christie, 2005). Its one of the most abundant species from Accipitridae in the neotropics. It also has a broad distribution that ranges from northern Mexico to Central Argentina including Brazil (Barros et al., 2010; Salcedo-Rivera et al., 2019). In Mexico, it is found along the Pacific coast, ranging from Jalisco to Chiapas, and from Tamaulipas to the Yucatán Peninsula (Bierregaard, Boesman & Kirwan, 2024). This species inhabits forest edges, open forests, disturbed areas in secondary forests, savannas with gallery forests, and scattered forests; it may even be present in urban habitats (Bierregaard, Boesman & Kirwan, 2024; Whitacre, Harris & Cordova, 1992; Whitacre et al., 1992). Their breeding season extends from late March to late July (Bierregaard, Boesman & Kirwan, 2024). The morphological traits of the studied species and their food preferences are summarized in Table S1.

Diet composition of raptors

We determined the diet of four raptor species in Mexico using two sources of information (1) photographic records available through the citizen science platforms iNaturalist (https://www.inaturalist.org/) and Macaulay Library (https://www.macaulaylibrary.org/) and (2) published records from articles or notes obtained through a scientific literature search. We searched both citizen science platforms for all available photographic records of the four studied raptors. Each photographic record was carefully reviewed and had to meet the following criteria to be included in the research: (1) the photograph had to be of sufficient quality to be able to identify with certainty the species of raptor, (2) the raptor had to be consuming the prey or holding it with its talons (perching or flying) (3) the remains of the prey had to evidence the biological group to which the prey belonged (taxonomic class when preys were vertebrates and subphylum when preys were invertebrates). This means the photograph had to show a diagnostic feature that would allow us to identify the prey with certainty. Examples of such features include hair, feathers, scales, fins, and exoskeletons. Finally, the (4) information on the geographic location and the date on which the record occurred had to be available. Additionally, we check for duplicity in the photographic records by different users or by the same user in both citizen science platforms; in these cases only one record was considered in our research.

In addition, we conducted a search for scientific literature that reported diet composition or prey consumption in Mexico by any of the four raptor species studied as part of their results. The search was conducted using the following repositories: Scientific Electronic Library Online (SciELO), Searchable Ornithological Research Archive (SORA), Google Scholar and Web of Science. In this search we used the following keywords and their combination in English and Spanish: Astur cooperii, Accipiter cooperii, Buteogallus anthracinus, Rupornis magnirostris, Buteo magnirostris, Buteo plagiatus, Buteo nitidus, diet, prey, trophic niche, and Mexico. We used search operators, or special characters and commands, to include exact phrases (“·”) or exclude words (−). This allowed us to refine our search and limit the articles to those with data from locations within the scope of our research. We reviewed each article obtained from these repositories, and only studies that met the following criteria were included: (1) the articles or notes had to include information about predatory interactions of the studied raptors. (2) The study site had to be within Mexico’s political boundaries. (3) The species or type of prey consumed by the raptor species had to be specified. (4) The geographical coordinates of where the predatory interaction occurred had to be provided. (5) The date of the predatory interaction had to be reported. Diet studies in raptors are usually based on the analysis of remains found in pellets. In these cases we only consider the presence of the prey in the diet of the raptor, i.e., we consider only one predation record regardless of the number of remains found in the pellets. On the other hand, from the photographic records and published records we extracted the following information: (1) species of the raptor that consumed the prey, (2) species of the prey or the lowest recognizable taxonomic grade to which the prey belonged or reported by the authors of the published records, (3) month and year in which the record occurred, and (4) geographic coordinates in which the record occurred. The search in both sources of information (citizen science platforms and scientific literature) was limited to records prior to January 1, 2024.

Records of predation in urban and non-urban areas

We identified which records of predation by raptors occurred in urban and non-urban areas. To achieve this, we projected the geographical coordinates (spatial point) of each predation event (as recorded in photographs and published records) onto a map of Mexico’s political boundaries using a geographic information system (ESRI, 2011). Subsequently, we projected a layer of information onto this map with Mexico’s land use categories (http://geoportal.conabio.gob.mx/metadatos/doc/html/usv250s7gw.html). This layer contains vegetation types and land uses across the entire country. For this research, we selected urban areas and human settlements from this layer, creating a new layer containing only these two land use categories. Urban areas and human settlements are defined as populated areas with a concentration of impervious surfaces (Instituto Nacional de Estadística y Geografía, 2023). We used the polygons of urban areas and human settlements (urban areas hereafter) as reference areas to identify records of predation by raptors that occurred within urban areas, while records outside these polygons were classified as records in non-urban areas. In addition, to verify that our dataset was not biased toward records close to urban areas, we calculated the distance of each predation record from the nearest human settlement (Tables S2 and Fig. S1). We projected all spatial points and polygons used to create all maps using the World Geodetic System 1984 (WGS84) geographic coordinate reference system and the Lambert conformal conic projection (ESRI, 2011).

Additionally, to determine whether the level of specialization in the diet of raptors changes in urban or non-urban areas we calculated standardized trophic niche breadth (Levins, 1968; Colwell & Futuyma, 1971; Hurlbert, 1978) using the following formula: Bi=1/(n−1){1∑jPij2−1} where Bi = standardized index of niche breadth, Pij proportion of diet of raptor i on type of prey j, and n = total number of prey types consumed by raptor i. This metric takes values from 0 to 1. Species with trophic niche breadth values less than 0.30 are classified as specialists, species with intermediate values (0.30–0.60) are classified as moderate specialists, and species with a generalist diet will have values greater than 0.60 (Krebs, 1999). We calculated trophic niche breadth using the spaa package (Zhang, 2013) in R v. 4.2.2 software (R Core Team, 2022).

Data evaluation

We estimated the sample coverage (SC) of the pairwise predation interactions between raptor species and prey (types and species of prey). Sample coverage is a measure of sample completeness and represents the proportion of the statistical population captured by the recorded predation interactions (photographs and published records), and is expressed as a function of the total number of records, singletons (pairwise predation interactions between raptor species and prey represented by exactly one record in the sample) and doubletons (pairwise predation interactions between raptor species and prey represented by exactly two records in the sample, Chao & Jost, 2012). Sample coverage takes values from 0 to 1, the maximum sampling completeness is reached with values of 1 (100% completeness); in other words, increasing the sampling effort (through photographs or published records) results in few or no new pairwise predation interaction between raptor species and prey (Chao & Jost, 2012). Sample coverage was calculated using iNext software (Chao, Ma & Hsieh, 2016). Additionally, we estimated the geographic coverage of our dataset. First, we project a grid with cells of 0.5° latitude by 0.5° longitude covering the total extent of Mexico (i.e., 794 cells) using a geographic information system (ESRI, 2011). We then projected the potential distribution of each of the raptors over this grid. We use the following polygons of potential distribution for raptor species: Cooper’s Hawk (Navarro-Sigüenza et al., 2018a), Common Black Hawk (Navarro & Peterson, 2007a), Roadside Hawk (Navarro-Sigüenza et al., 2018b) and Gray Hawk (Navarro & Peterson, 2007b). Then, we calculated the total number of cells covered by each raptor’s potential distribution area. Subsequently, we projected the geographical coordinates (spatial point) of each predation event over this map. Finally, we calculated the number of predation records for each raptor species within cell to determine the geographic coverage of the records and identify areas of their distribution lacking information.

Statistical analysis

To determine whether raptors had a preference for any particular prey we used a chi-square test and post hoc pair-wise comparisons (Cox & Key, 1993). The expected values were an equal proportion of consumption among all prey types, while the observed values were the proportion of all prey types consumed obtained from citizen science data and literature. To compare the number of records of predation by raptors occurring in urban and non-urban areas, we used a Mann–Whitney U test. To determine whether raptors changed their proportion of prey types consumed in urban and non-urban areas, we used a chi-square test. The expected values were the proportions of prey consumed in non-urban areas, while the observed values were the proportion of prey consumed in urban areas. Finally, we used GLMs (Poisson distribution, log link function) to test whether prey types, months of year and breeding season influenced the frequency of predation records by raptors. In each model, we include the type of prey most consumed, the month with the highest number of predation records and breeding season of each raptor species as reference variables (i.e., a reference level within a category that enables the interpretation of the other levels in relation to it). We excluded records of predation from these models in which the month of observation was not reported (Table S2). The number of records included in each model for each species is as follows: Cooper’s Hawk = 123, Common Black Hawk = 107, Roadside Hawk = 84 and Gray Hawk = 52. We reported the Akaike information criterion and the proportion of the deviance explained by each full model (R2). Finally, since our data were mainly extracted from photographs available on citizen science platforms, the frequency of predation records may be related to the number of records available on these platforms. So, we used a Spearman’s rank correlation to test whether the monthly records of the raptors in citizen science platforms were correlated with the frequency of monthly predation records by raptor species.

Results

We reviewed 27,923 photographic records, of which 18,717 records (67%) were reviewed in iNaturalist and 9,206 (33%) in Macaulay Library. The vast majority of these photographic records showed raptor species perching or flying. Our research did not include these photographic records because they did not show evidence of predatory interaction. Four photographic records could not be identified at a high taxonomic level, so they did not meet the criteria for inclusion in this study (Table S2). Only 356 (1%) photographic records met the criteria for inclusion in this research. These records covered a time period from January 2000 to December 2023. The number of records in iNaturalist and Macaulay Library by species included in this research was as follows: Cooper’s Hawk = 122 (34%), Common Black Hawk = 106 (30%), Roadside Hawk = 80 (23%) and Gray Hawk = 48 (13%). The literature review provided 45 predation records through 10 articles (Table S3). The number of records reported in the literature for each species was as follows: Cooper’s Hawk = 30 (67%), Common Black Hawk = 4 (9%), Roadside Hawk = 5 (11%) and Gray Hawk = 6 (13%). By gathering this information for the four raptor species from both sources of information, we obtained 401 predation records of 91 prey species in Mexico (Table S2 and S3). The sample coverage of predation interactions between raptors and their prey types had high levels of completeness for the four raptor species (Cooper’s Hawk = 1, Common Black Hawk = 0.99, Roadside Hawk = 0.97 and Gray Hawk = 0.98). On the other hand, the completeness of the recorded prey species was lower than the level of the type of prey (Cooper’s Hawk = 0.77, Common Black Hawk = 0.86, Roadside Hawk = 0.74 and Gray Hawk = 0.72). The geographic coverage of predation event records was variable among raptor species (Fig. 1, Table 1). Cooper’s Hawk was the species that had the least geographic coverage with predation records in 8% of the cells that covered its potential distribution (Fig. 1A, Table 1). On the other hand, Roadside Hawk was the species that had the greatest geographic coverage with predation records in 14% of the cells that covered its potential distribution in Mexico (Fig. 1C, Table 1). Some cells outside the distribution potential range of the species had predation records as follows: Cooper’s Hawk = 1 cell, Common Black Hawk = 3 cells, Roadside Hawk = 2 cells and Gray Hawk = 1 cell (Fig. 1).

Figure 1 Geographical coverage maps of prey consumption records of four raptors in Mexico.

(A) Cooper’s Hawk, (B) Common Black Hawk, (C) Roadside Hawk and (D) Gray Hawk. Darker colored cells indicate a higher number of predation records. White cells indicate the absence of predation records. The area covered by the diagonal grid is the potential distribution of raptor species in Mexico.

Table 1 Geographical coverage of predation events of four species of raptors in Mexico.

The number of cells that covered the geographic distribution of each species is indicated, as well as the number of cells with predation events and the percentage of cells with predation events.

Raptor species	Potential distribution (cells)	Cells with records of predation events (n)	Percentage of cells with records of predation events (%)	
Cooper’s Hawk	723	58	8.02	
Common Black Hawk	409	43	10.51	
Roadside Hawk	227	33	14.54	
Gray Hawk	430	41	9.53	

The raptors studied consumed prey from all five vertebrate classes and three invertebrate subphyla (Fig. 2). The raptors consumed prey types in a proportion that was significantly different from what was expected (Cooper’s Hawk, Χ2 = 210.57, df = 2, P < 0.0001; Common Black Hawk, Χ2 = 113.75, df = 6, P < 0.0001; Roadside Hawk, Χ2 = 139.41, df = 6, P < 0.0001 and Gray Hawk, Χ2 = 33.59, df = 4, P < 0.0001). All raptors had a preferred type of prey; that is, we found significantly more records of the consumption of that prey type than the other prey types: Cooper’s Hawk consumed significantly more birds than other prey (mammals P < 0.0001; reptiles P < 0.0001). Common Black Hawk consumed similar proportions of fishes and crustaceans (P = 0.05) but consumed significantly more fishes than other prey (amphibians, P < 0.0001; birds P < 0.0001; insects = P < 0.0001; mammals P < 0.0001; reptiles P = 0.007). Gray Hawk consumed similar proportions of reptiles and birds (P = 0.71), reptiles and mammals (P = 0.49) but consumed significantly more reptiles than amphibians (P < 0.001) and crustaceans (P < 0.001). Roadside Hawk consumed significantly more reptiles than other prey (amphibians P < 0.0001; birds P < 0.0001; crustaceans P < 0.0001; insects P < 0.0001; mammals P < 0.0001 and myriapods P < 0.0001; Table S3, Fig. 2).

Figure 2 Diet composition of four raptors in Mexico.

Bars are the proportion of types of prey consumed by each raptor. The color of bars indicates each type of prey in the diet. The numbers to the right of each bar indicate the percentage of each type of prey in the raptor’s diet.

The distance between raptor predation records and the nearest human settlement ranged from 0 to 36.4 km (Cooper’s Hawk, the range was 0 to 25.6 km; Gray Hawk, 0 to 16.9 km; Common Black Hawk, 0 to 28 km; and for Roadside Hawk, 0 to 36.4 km; Table S2 and Fig. S1). Most records of predation by Common Black Hawk (urban areas = 23 records, non-urban areas = 87 records; Fig. 3B), Roadside Hawk (urban areas = 28 records, non-urban areas = 57 records; Fig. 3C), and Gray Hawk (urban areas = 18 records, non-urban areas = 36 records; Fig. 3D) occurred in non-urban areas. On the contrary, most records of predation by Cooper’s hawks occurred within urban areas (urban areas = 79 records, non-urban areas = 73 records; Fig. 3A). The number of predation records by Cooper’s Hawk was higher in urban areas than in non-urban areas (Mann–Whitney = 33.5, z = 2.21 (non-urban = 73, urban = 79), P = 0.024; Fig. 4A). In contrast, the number of predation records was higher in non-urban areas than in urban areas for Common Black Hawk (Mann–Whitney = 15, z = 3.28 (non-urban = 87, urban = 23), P < 0.001; Fig. 4B), Roadside Hawk (Mann–Whitney = 34.5, z = 2.15, (non-urban = 57, urban = 28), P = 0.029; Fig. 4C) and Gray Hawk (Mann–Whitney = 38, z = 2.03, (non-urban = 36, urban = 18), P = 0.041; Fig. 4D). The richness of prey types consumed by Cooper’s Hawk and Gray Hawk remained constant in urban and non-urban areas (Figs. 5A and 5D). In contrast, fewer prey types were consumed by Common Black Hawks and Roadside Hawks in urban areas than in non-urban areas (Figs. 5B and 5C). The proportion of prey types consumed in urban and non-urban areas changed significantly for Cooper’s Hawk (Χ2 = 15.6, df = 2, P = 0.0004; Fig. 5A), Roadside Hawk (Χ2 = 21.75, df = 6, P = 0.0005; Fig. 5C), and Gray Hawk (Χ2 = 56.07, df = 4, P < 0.0001; Fig. 5D). Conversely, the proportion of prey types consumed by Common Black Hawk’s was similar in urban and non-urban areas (Χ2 = 5.83, df = 6, P = 0.46; Fig. 5B). The trophic niche breadth of Common Black Hawk (urban areas = 0.67, non-urban areas = 0.39), Roadside Hawk (urban areas = 0.44, non-urban areas = 0.24), and Gray Hawk (urban areas = 0.60, non-urban areas = 0.48) was broader in urban areas than in non-urban areas. In contrast, Cooper’s Hawk had a narrow trophic niche breadth in urban areas than in non-urban areas (urban areas = 0.03, non-urban areas = 0.27).

Figure 3 Location of predation records by raptor in urban and non-urban areas in Mexico.

(A) Cooper’s Hawk, (B) Common Black Hawk, (C) Roadside Hawk and (D) Gray Hawk. Urban areas = black, non-urban areas = gray, predation records in urban areas = pink squares, predation records in non-urban areas = blue triangles.

Figure 4 Number of predation records by raptors in urban and non-urban areas in Mexico.

(A) Cooper’s Hawk, (B) Common Black Hawk, (C) Roadside Hawk and (D) Gray Hawk. The boxes represent 50% of the data divided by the median of each sample (horizontal line), the vertical lines indicate the limits of 0.95 and 0.05. The gray dots are the monthly records of predation by each species in urban and non-urban areas. The black dots are outliers from each distribution.

Figure 5 Comparison between the diet composition of four raptors in urban and non-urban areas of Mexico.

Bars are the percentage of preys consumed by each species. (A) Cooper’s Hawk, (B) Common Black Hawk, (C) Roadside Hawk and (D) Gray Hawk. The color of bars indicates each type of prey in the diet. The numbers to the right of each bar indicate the percentage of each type of prey in the raptor’s diet.

GLMs indicated that the frequencies of predation records by raptors were differentially associated with the month of the year, the type of prey consumed and breeding season (Table 2, Fig. 6). The highest frequency of predation records by Roadside Hawk occurred during their breeding season (Table 2, Fig. 6C). In contrast, the highest frequency of predation records by Cooper’s Hawks occurred during their non-breeding season (Table 2, Fig. 6A). On the other hand, the frequency of predation records by Common Black Hawks and Gray Hawk was independent of their breeding season (Table 2, Figs. 6B and 6D). Monthly records of Common Black Hawk on citizen science platforms were correlated with monthly predation records (Spearman r = 0.71, N = 12, P = 0.008; Fig. 6B). However, there was no correlation between monthly records on citizen science platforms and monthly predation records by Cooper’s Hawk (Spearman r = 0.53, N = 12, P = 0.07; Fig. 6A), Roadside Hawk (Spearman r = −0.22, N = 12, P = 0.48; Fig. 6C) and Gray Hawk (Spearman r = −0.26, N = 12, P = 0.40; Fig. 6D).

Table 2 Summary of generalized linear models.

The GLMs evaluated whether the frequencies of predation records were associated with the month of the year, the type of prey consumed by raptor species and the breeding season. In each model, we include the type of prey most consumed, the month with the highest number of predation records by each raptor and breeding season (months) as reference variables as follows: Cooper’s Hawk = birds, January, March to July; Common Black Hawk = fishes, January, April to August; Roadside Hawk = reptiles, April, March to July; and Gray Hawk = reptiles, June, April to August. Only months with a frequency of predation records significantly different from the month used as a reference variable are shown. Values in bold indicate significant differences from the reference variables.

	Estimate	ES	Z	P	
Model 1 = Cooper’s Hawk, AIC = 91.31, R2 = 0.97	
Intercept	2.77	0.25	11.09	<0.001	
April	−1.16	0.51	−2.27	0.023	
May	−1.12	0.48	−2.32	0.020	
August	−1.16	0.51	−2.27	0.023	
Mammals	−1.88	0.47	−3.93	<0.001	
Reptiles	−2.07	0.61	−3.39	<0.001	
Not breeding	1.12	0.48	2.32	0.020	
Model 2 = Common Black Hawk, AIC = 158.20, R2 = 0.69			
Intercept	1.94	0.27	7.04	<0.001	
April	−0.95	0.45	−2.08	0.036	
June	−1.46	0.63	−2.31	0.020	
Amphibians	−1.18	0.45	−2.62	0.008	
Aves	−1.11	0.62	−1.77	0.076	
Mammals	−1.94	1.03	−1.87	0.060	
Reptiles	−0.83	0.28	−2.92	0.003	
Crustaceans	−0.16	0.24	−0.68	0.492	
Insects	−1.46	0.75	−1.94	0.052	
Not breeding	0.40	0.22	1.79	0.072	
Model 3 = Roadside Hawk, AIC = 132.16, R2 = 0.82			
Intercept	2.27	0.27	8.30	<0.001	
January	−1.20	0.52	−2.31	0.020	
June	−1.16	0.56	−2.08	0.037	
August	−1.53	0.63	−2.42	0.015	
October	−1.91	0.76	−2.50	0.012	
December	−1.73	0.65	−2.66	0.007	
Amphibians	−1.14	0.50	−2.27	0.022	
Aves	−0.86	0.32	−2.63	0.008	
Mammals	−1.04	0.43	−2.42	0.015	
Crustaceans	−2.27	1.03	−2.19	0.028	
Insects	−0.93	0.40	−2.27	0.023	
Myriapods	−1.62	1.05	−1.52	0.123	
Not breeding	−0.50	0.22	−2.26	0.023	
Model 4 = Gray Hawk, AIC = 111.76, R2 = 0.46			
Intercept	1.45	0.39	3.68	<0.001	
Amphibians	−0.80	1.13	−0.71	0.477	
Aves	−0.13	0.36	−0.36	0.716	
Mammals	−0.23	0.37	−0.61	0.538	
Crustaceans	−0.62	1.11	−0.56	0.571	
Not breeding	−0.11	0.27	−0.41	0.676	

Figure 6 Monthly predation records by raptors.

(A) Cooper’s Hawk, (B) Common Black Hawk, (C) Roadside Hawk and (D) Gray Hawk. The color of bars indicates each type of prey in the diet. The numbers to the right of each bar indicate the percentage of each type of prey in the raptor’s diet. The dotted line and secondary y-axis are the monthly number of sightings recorded for each species of bird of prey. The black line below the x-axis is the length of the breeding season for each species of raptor.

Discussion

Research into the diet of raptors has important implications for their conservation, as it affects the maintenance of their populations, their health, and even the population dynamics of their prey (Boal, 2018; McClure et al., 2018). However, little attention has been paid to the diet composition of raptors inhabiting urban areas. In this study, we determined the diet composition of four raptors and compared whether it changed in urban and non-urban areas of Mexico. To achieve this, we created a dataset using data extracted from a literature review and photos available on citizen science platforms. Our dataset of predation interactions between raptors and their prey types had high range of sample coverage, from 0.97 to 1. This means that the information extracted from photographs on citizen science platforms allowed us to determine with a high degree of certainty the composition of the diet of raptors in terms of the types of prey consumed (Fig. 2). In addition, this high coverage of the dataset allows us to infer the preferences of raptors for preying on certain types of prey. Cooper’s Hawk consumed significantly more birds than other prey. This is consistent with what was reported in other locations throughout its the geographical distribution (Kennedy & Johnson, 1986; Hiraldo et al., 1991; Roth & Lima, 2003; Rosenfield et al., 2024). The Common Black Hawk showed a clear preference for preying on crustaceans and fishes; both types of prey have been reported as the most commonly consumed prey by this species (Johnson & Schnell, 2024). Roadside Hawk consumed significantly more reptiles than other prey. The diet of this species is reported to be generalist and highly opportunistic, and its preference for preying on certain types of prey varies widely throughout its geographical distribution (Bierregaard, Boesman & Kirwan, 2024). However, reptiles often compose an important part of their diet (Panasci & Whitacre, 2000). Gray Hawk did not have a clear preference for preying on one type of prey; this species preyed on reptiles, birds, and mammals in similar proportions. Our results are consistent with previous reports, which indicate that the species’ diet consists mainly of vertebrates, particularly reptiles, birds, and mammals (Bibles, Glinski & Johnson, 2020). On the other hand, the sample coverage at the species level of prey consumed by raptors was a range of sample coverage from 0.72 to 0.86. These results highlight the effectiveness of our dataset in identifying prey even at low taxonomic levels (see Table S2) and the complexity of the diet of the studied raptors and their great ability to include opportunistically new prey items in their diet (Jaksić & Braker, 1983).

The geographic coverage of predation records for the four raptor species was low. Only 8–14% of the cells that covered the potential distribution of raptors in Mexico had records of predation events (Fig. 1, Table 1). However, the cells with predation records were dispersed latitudinally and longitudinally over most of the potential geographic range of the four raptor species, i.e., the records covered a wide variety of localities facilitating the recording of greater prey richness. Also, we were able to detect regions of the country with a lack of records of predation events. Particularly, in the northwestern of the country (Sonora and Sinaloa) we found no records of Cooper’s Hawk predation events. Also, the coasts of the states of Guerrero and Michoacán lack records of consumed prey by Gray Hawk and Common Black Hawk. In the case of the Roadside Hawk, the absence of records extends to the coasts of Guerrero, Michoacán, Colima, and Jalisco. This highlights the importance of conducting research in these locations, as well as promoting social participation in citizen science platforms in these regions to expand knowledge of these species diets and their local dietary preferences.

Although most of our data comes from citizen science platforms, our dataset did not show a spatial bias toward urban areas (see Table S2 and Fig. S1). Therefore, our results accurately reflect the diets of the studied raptors and their dietary shifts in urban areas in Mexico. Contrary to our expectations, the richness of prey types consumed by Cooper’s Hawk and Gray Hawk remained constant in urban and non-urban areas. The diet of both species is based on the consumption of reptiles, birds, and mammals (Figs. 2, 5A, and 5D). Some species of these types of prey are characterized by being human-commensals and respond favorably to urbanization such as pigeons, rats, and even some lizards (Chace & Walsh, 2006). These species can take advantage of direct food supplementation through bird feeders, waste generated by the human population or artificial lights that attract and concentrate high abundance of invertebrates (Chace & Walsh, 2006; Putman & Tippie, 2020). The abundance and availability of these prey within cities allowed the richness of prey types consumed by both raptors to remain stable in urban areas. In contrast, and as we expected, the richness of prey types consumed by Common Black Hawk and Roadside Hawk was lower in urban areas than in non-urban areas. Both raptors consumed seven types of prey in non-urban areas and only five types of prey in urban areas (Figs. 5B and 5C). The types of prey that were not consumed in urban areas by these raptors were one class of vertebrates (mammals) and three subphyla of invertebrates (crustaceans, myriapods, and insects). Urbanization acts as an ecological filter that promotes a reduction in species richness and biotic homogenization through the increase in the abundance of generalist species that exploit resources and conditions within urban areas (McKinney, 2008). These changes in the richness and abundance of available prey lead raptors to shift the composition of their diet to abundant prey (Boal, 2018; Leveau et al., 2022). However, the decrease in the richness of prey types consumed by Common Black Hawk and Roadside Hawk in urban areas did not significantly alter the diet composition of these species, as these prey had very low proportion of consumption in non-urban areas (1–2% of the proportion of prey consumed, Figs. 5B and 5C).

As we expected, the proportion of prey types consumed by Cooper’s Hawk, Roadside Hawk, and Gray Hawk was significantly different in urban areas than in non-urban areas (Figs. 5A, 5C, and 5D). In contrast, the proportion of prey types consumed by Common Black Hawk was similar in urban and non-urban areas (Fig. 5B). Regardless of their preference for consuming a particular type of prey, the four raptors studied increased the proportion of bird consumption in urban areas (Fig. 5). The increase in the proportion of bird consumption was from 2% to 36% (Fig. 5), with the Common Black Hawk showing the smallest increase in bird consumption (2%) and the Gray Hawk showing the highest increase in bird consumption in urban areas (36%). This pattern is consistent with previous reports indicating that some species raptors increase the proportion of birds they consume in urban areas, while reducing the proportion of other types of prey, such as mammals or invertebrates (Ballejo & De Santis, 2013; Sumasgutner et al., 2014). The increase in bird consumption in urban areas could be related to the following factors: the high abundance of some exotic species such as pigeons (Columba livia), Starlings (Sturnus vulgaris) or sparrows (Passer domesticus, Jaksić & Braker, 1983; Cava, Stewart & Rosenfield, 2012), the constant food supplies lead to the increase in the biomass of some bird species (Chace & Walsh, 2006), and the ease of hunting birds that are attracted to artificial feeders (McCabe et al., 2018). An important component in the diet of raptors in urban areas is the consumption of exotic bird species (Cava, Stewart & Rosenfield, 2012). This implies that the establishment and persistence of raptor populations in urban areas could help regulate the populations of exotic bird species. However, it has been reported that some raptors tend to have higher rates of Trichonomiasis in urban areas than in non-urban areas due to the consumption of infected birds (Boal & Mannan, 1999). The prevalence of the disease could be higher in urban areas due to the use of feeders by infected birds. The increase in the proportion of bird consumption in urban areas could increase the exposure of raptors to the disease and lead to a higher mortality rate, affecting the conservation of their populations. This highlights the need for research focused on assessing the prevalence of Trichomoniasis in urban bird and raptor populations. On the other hand, the proportion of prey consumed by Common Black Hawk was similar in urban and non-urban areas. This can be explained by the strong preference of this raptor for habitats associated with water bodies (Johnson & Schnell, 2024). These habitats are associated with the availability and abundance of their preferred prey (fishes and crustaceans; Figs. 2 and 5B). This suggests that this species will only occur in human settlements that meet these habitat and food requirements, which would imply areas with low or intermediate levels of urbanization. This is consistent with the findings of Boal (2018), who reports that raptor species most closely associated with urban areas will have a broad food-niche and a preference for more closed habitats such as forests.

As we expected the trophic niche breadth of Common Black Hawk, Roadside Hawk and Gray Hawk was broader in urban areas than in non-urban areas. Boal (2018) reports that trophic niche breadth of raptors tends to be broader in urban areas. In this study the raptors maintained or consumed fewer prey types in urban areas. In addition, all studied raptors increased the proportion of birds consumed in urban areas. The trophic niche breadth is determined by the availability of local resources (Jaksić & Braker, 1983). In urban areas, the high availability of birds and their aggregation around feeders makes it easier for raptors to hunt them. These factors lead to a more equitable distribution of prey consumption in urban areas and a more generalized diet. Contrary to our expectations, Cooper’s Hawk had a narrow trophic niche breadth in urban areas than in non-urban areas. Cooper’s Hawk has a diet specialized in consuming birds. The abundance and constant availability of birds in urban areas allows Cooper’s Hawk to consume more birds in urban areas than in non-urban areas.

Temporal variation in the frequency of predation records was influenced by differential factors depending on the species of raptor. The monthly proportion of consumption of different types of prey varied throughout the year. This variability could highlight changes in energy requirements mainly associated with the breeding season, particularly egg laying and chick rearing, as their high growth rate demands more energy (Catry et al., 2016). Similarly, these seasonal changes in diet could reflect the natural dynamics of prey abundance and availability in the environment (Cava, Stewart & Rosenfield, 2012; Garcia-Heras et al., 2017). Only Roadside Hawk met our expectations, as the frequency of its predation records increased during the months of its breeding season (Table 2, Fig. 6C). It has been reported that in this species, the prey delivery rate to chicks in the nest ranges from 0.26 to 0.43 prey/hour/chick (Panasci & Whitacre, 2002). This implies constant activity around the nest generated by continuous provisioning by both parents. This increased activity in a specific location (within the territory and at the nest) could facilitate the photographic recording of these predation events. In contrast, Cooper’s Hawk showed the opposite pattern, as the highest frequency of its predation records occurred during the non-breeding season (Table 2, Fig. 6A). It has been reported that in avian predatory raptors, the size of their prey decreased during their breeding season (Newton & Marquiss, 1982; Panter & Amar, 2022). The coincidence of the breeding season of these raptors and the emergence of passerine fledglings could explain this change in prey size selection (Newton & Marquiss, 1982). The main source of our data was photographic records on citizen science platforms. This source of information is biased toward the recording of predation on large prey, as their identification at lower taxonomic levels is usually possible and their handling time is longer, which facilitates the photographic recording of the predation event (see study limitations). The change in prey size selection during the breeding season of Cooper’s Hawks may have limited the number of predation event records, resulting in a higher number of predation event records during the non-breeding season. On the other hand, the monthly variation in Common Black Hawk predation records showed a correlation with the number of monthly photographic records on citizen science platforms. The activity of observers who upload their photographic records to citizen science platforms can vary throughout the year. Holiday seasons or initiatives such as Global Big Day can generate a temporary increase in photographic records on these platforms. Research based on citizen science records should consider this possibility, especially when evaluating patterns of frequency of records, since the activity of users of these platforms could be a factor that obscures the ecological patterns being evaluated. The variation in Gray Hawk’s monthly predation records did not show a pattern associated with prey type, breeding season, or citizen science platform presence records. The low number of predation records for this species may have prevented us from detecting a clear pattern that we can associate with the temporal changes in its diet. On the other hand, the generalist and opportunistic diet of the species (Bibles, Glinski & Johnson, 2020) may prevent us from detecting a clear pattern in the monthly variability of its diet. In particular, Gray Hawk and Common Black Hawk had low numbers of predation events within urban areas. This can be explained by their association with certain habitat characteristics (forested areas and bodies of water, respectively) or by their preferred prey (reptiles, and fish, and crustaceans, respectively). The low number of predation events in urban areas may have prevented us from identifying temporal patterns related to the breeding season. Future research could focus on evaluating how the reproductive activity of both species promotes temporal variability in predation records.

Study limitations

The compilation and analysis of information extracted from photographic records available on citizen science platforms has proven to be an effective tool for determining the dietary patterns of raptors at larger temporal and spatial scales than using traditional approaches (Naude et al., 2019; Panter & Amar, 2022; Panter et al., 2024). However, biases have been detected associated with the use of these data. For example, prey were sometimes impossible to identify even at high taxonomic levels; in the present study, we were unable to identify the prey at a high taxonomic level in only four photographic records (<1%; Table S2). This could be explained by the fact that the studied raptors are large enough to hunt larger prey, which facilitates photographic recording of predation and subsequent prey identification. This should be taken into account when conducting research on raptors that prefer smaller prey. Thus, the probability of successfully identifying prey at low taxonomic levels depends on its consumption level, so larger prey or preys that require longer handling times (e.g., mammals, birds, or crustaceans) will be more easily photographed and identified. This implies, that predation records of small prey (e.g., arthropods) that are consumed quickly may not be recorded, generating a bias towards larger prey (Panter & Amar, 2022; Panter et al., 2024). On the other hand, our data mainly comes from citizen science photographs. The use of these platforms has become widespread and their usefulness in scientific research has been widely demonstrated. However, only 1% of the photographic records reviewed met the criteria for inclusion in this research. This highlights that this approach is useful for abundant species, species with a wide geographic distribution or highly photographed charismatic species. Therefore, different approaches must be used for the study of rare or cryptic species. For example, the use of camera traps in raptor nests to monitor the types of prey that adults bring to their chicks (McPherson, Brown & Downs, 2016) or use molecular techniques such as metabarcoding to identify the remains of prey from the surface of raptor beaks and talons (Bourbour et al., 2021).

Conclusions

The analysis of information obtained from photographic records available on citizen science platforms, as well as the compilation of data published in the scientific literature, allowed us to determine with a high degree of certainty the diet composition of the four raptor species on a broad geographic scale. The effect of urban areas on the richness and proportion of prey consumed by raptors differed among the species studied. The richness of prey types consumed by Cooper’s Hawk and Gray Hawk remained constant in urban and non-urban areas. However, the richness of prey types consumed by Common Black Hawk and Roadside Hawk was lower in urban areas than in non-urban areas. The proportion of prey types consumed by Cooper’s Hawk, Roadside Hawk, and Gray Hawk was significantly different in urban areas than in non-urban areas. In contrast, the proportion of prey types consumed by Common Black Hawk was similar in urban and non-urban areas. The trophic niche breadth of raptors was broader in urban areas than in non-urban areas, except for Cooper’s hawks, which had a narrow trophic niche breadth in urban areas than in non-urban areas. This is associated with Cooper’s Hawk preference for feed on birds. The four raptors studied increased the proportion of bird consumption in urban areas. This change in the proportion of bird consumption in urban areas could imply a risk to the conservation of raptor populations due to the higher prevalence of diseases transmitted through the consumption of infected birds. This highlights the need for research focused on assessing the prevalence of diseases that pose a risk to the conservation of birds and raptors in urban areas.

Supplemental Information

Supplemental Information 1 The morphological traits of the studied raptors and their prey types.

All types of prey reported for each species of raptor are indicated. The ranges in the table include data from males and females. The references from which the data were obtained are indicated in parentheses: a = Schnell 1994; b = Wheeler & Clark 1995; c = Fergusson-Lees Christie, 2005; d = Cartron et al., 2010; e = Duffy, 2012; f = Garay & Marín, 2019; g = Bibles et al., 2020; h = Stewart et al., 2023; i = Bierregaard et al., 2024; j = Johnson & Schnell, 2024; k = Rosenfield et al., 2024.

Supplemental Information 2 Raw data from records of prey consumed by studied raptors.

The species of the prey, the taxonomic class when preys were vertebrates and in subphylum when preys were invertebrates, the information on the geographic location and the date on which the record occurred, the area type classification as well as the nearest distance to human settlements and the source of information (link to record of citizen science platform or the literature reference) are shown. The last four records, marked with an asterisk (*), were excluded from the analysis because they did not meet the inclusion criteria for this study.

Supplemental Information 3 Prey consumption records of four species of raptors in Mexico.

Records were compiled through two sources: citizen science platforms (CS) and a review of published articles (L). Records associated with lowercase letters were obtained from the following publications: a = Hiraldo et al., 1991; b = Ibarra-Zimbrón et al., 2001; c = Mikula et al., 2015; d = Rodríguez-Canseco et al., 2015; e = Nahuat-Cervera et al., 2020; f = Ortega-Álvarez et al., 2022; g = Bello-Sánchez et al. 2021; h = Valencia-Herveth et al., 2022; i = Sánchez et al., 2023; and j = Escamilla-Cortés & García-Grajales, 2024.

Supplemental Information 4 Distance from each predation record to the nearest human settlement.

We calculated the distance from each predation record to the nearest human settlement using a geographic information system (ESRI, 2011). The distance of records that occurred within the polygons of human settlements was “0”. These records were classified as urban records, while records that occurred outside the polygons of human settlements were categorized as non-urban

We thank the three anonymous reviewers for providing useful comments that greatly improved an early version of the manuscript. We want to thank to the citizen science platforms iNaturalist (https://mexico.inaturalist.org/) and eBird/Macaulay Library (https://ebird.org/region/MX, https://search.macaulaylibrary.org/catalog?regionCode=MX) and their users, without whom the present study wouldn’t be possible. Dedicated to the beloved memory of Yolotli and to my beloved father Carlitos. Thank you for being an important part of my life and who I am.

Additional Information and Declarations

Competing Interests

The authors declare that they have no competing interests.

Author Contributions

Ubaldo Márquez-Luna conceived and designed the experiments, performed the experiments, analyzed the data, prepared figures and/or tables, authored or reviewed drafts of the article, and approved the final draft.

Oscar Rivera-Hernández performed the experiments, analyzed the data, authored or reviewed drafts of the article, and approved the final draft.

Data Availability

The following information was supplied regarding data availability:

The raw data is available in the Supplemental Files.

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
