# Peer review of "Urban areas promotes shifts in the proportion of prey consumed by four raptor species (Accipitridae) in Mexico"

_PeerJ, doi:10.7717/peerj.20307_

## Round 0.1 · original submission · Major Revisions

Thank you for submitting "Diet composition in four raptor species (Accipitridae) in Mexico". After review, we invite you to submit a major revision. The reviewers’ main requests are: an improved bibliographic review, detailed methods (statistics) and results, and a rewrite of much of its discussion and conclusions.

We look forward to receiving your improved manuscript.

Sincerely,
Best regards,

Armando Sunny

Reviewer 1 ·

Basic reporting

This study describes the diet of four species of raptors in Mexico, based on both the analysis of images from naturalistic platforms and the literature. The results are highly qualitative and show some geographic differences compared to the diets of northern studies.
I am not a native English speaker, so I cannot give an accurate opinion on the standards of the written language.
The introduction seems appropriate, with an adequate review of the literature; however, I consider that the review was biased toward descriptive studies of the diet, leaving out works focused on the ecology and natural history of the study species.
Figures and tables are appropriate although they need editing to improve clarity.

Experimental design

The original primary research is consistent with PeerJ's objectives and scope. The research question is defined, filling an identified knowledge gap. The research conducted requires clarification of some methodological details. A more rigorous analysis supported by inferential statistics is also necessary to demonstrate the apparent geographic differences in the species' diets. The dietary breadth indices used also need to be clarified.

Validity of the findings

Overall, the discussion is the weakest part of the paper because it requires a rethinking of the results and the clarification of some apparent contradictions. The conclusions are also speculative because one of the two objectives stated in the study was not fully developed.

Additional comments

Title: Considering the results obtained in this study, I suggest modifying the title: “Interspecific differences in diet composition in four raptor species (Accipitridae) in Mexico”

Abstract
Line 16-17: “lack studies about their diet composition and their prey preferences” … I find this sentence redundant, please correct it.
Introduction
Lines 85-88. It is necessary to justify why only these 4 species were chosen, out of the 38 present in Mexico.
Materials and methods
Lines 107-140. The description of the four study species was based solely on their morphology and distribution, but there is no information on their natural history. Considering that this study focuses on describing the species' diet, it is necessary to include a description that provides relevant information on their ecology.
Table 1. The table is confusing in its content, because the "prey preferences" section indicates that only prey items with the highest consumption rates (e.g., frequencies, percentages, etc.) are included. However, it appears that the table includes all prey items recorded in the birds' diets, regardless of their high or low consumption. My suggestion is, on the one hand, to clarify whether the table truly shows "preference" data or whether it includes all prey items in the diet. In the latter case, it would be helpful to highlight (e.g., in bold) those prey items with the highest consumption rates.
Lines 142-158. Please mention how many valid diet records were obtained online after the filtering, and how many records were based on the literature.
Lines 165-166. I have the impression that the bibliographic search focused only on those works where diet is the main feature; however, it seems that all works focused on the description of the ecology or natural history of the species, where diet is also included as part of the studies, were excluded. Consequently, the results may be incomplete because they omit more extensive works with descriptions of the species' diet.
Lines 181-183. Specify which was the source of consultation to obtain the potential distribution of the birds.
Lines 190-191. Clarify which index(es) were used to estimate the standardized trophic niche breadth, in addition to justifying their choice.
Results
Lines 196-205. The wording of the obtained records is a bit confusing, I suggest making the following changes: Line 198: Write: The number of records in iNaturalist and eBird by species included in this research was as follows: …..Line 201: Write: The number of records reported in the literature for each species was as follows:…..
Line 200: “The literature review provided 43 predation records through 8 articles”. "Predation records" can be considered a very different concept from "diet composition," which was the recorded variable described in the methods. I detected ambiguity in the records obtained, and therefore it is necessary to clarify specifically the types of records obtained and reported. On the other hand, all reported diet records based on feces (as mentioned in the introduction) would be outside this result based on "predation records"
Lines 200-201. Table 2 does not distinguish between records obtained in iNaturalist-eBird and the literature. It would also be convenient to include in table the totals for each column (species).
Lines 206.210. Only a qualitative description of interspecific differences in diet is included, however, I suggest including a statistical analysis to determine whether these dietary differences are significant.
One of the study's objectives was to determine the proportion of different types of prey consumed by raptors. However, this objective remains unfulfilled because it is only briefly reported in Figure 1. There is no detailed description or analysis of the proportions of prey ingested. I consider this analysis necessary.
Discussion
Line 223. It is highly speculative to claim that there are interspecific differences in diet when the analysis is based solely on a priori qualitative description. A statistical analysis is required to confirm these differences.
Lines 224-228. This description of the prey percentages for each bird species should be part of the results, to be later considered in the discussion.
Lines 228-234. These categories of dietary specialists and generalists should also be included in the results.
The authors state that the diet composition of raptors may change following latitudinal or longitudinal patterns throughout their geographic distribution, however, the study does not analyze this geographic variation within its results nor does it discuss the possibility of this variation. Were geographic differences in diet detected when comparing the study species in Mexico versus other latitudes-longitudes?
Lines 248-251. This sentence is very confusing: "Our results indicate that the Roadside Hawk consumed seven types of prey. However, in Mexico this species has a preference for consuming reptiles." The study results are based on records from Mexico (with seven recorded prey), yet in Mexico, they only consume reptiles? It's a contradiction.
Lines 251-256. Ok, But what would then be the conclusion about this latitudinal pattern of dietary variation? Can the species be considered opportunistic, consuming available prey without specific preferences?
Lines 273-274. But the latitudinal effect encompasses many possible proximate mechanisms determining diet, such as the spatial availability of prey, innate food preferences, possible intraspecific variations in foraging habits, among many other possible explanations. I think the claim that the "latitudinal effect could explain the specialized reptile diet of the Roadside Hawk in Mexico" is very superficial because it encompasses many possible explanations.
Lines 302-305. A contradiction of this apparent usefulness is raised later when you state that "only 1% of the photographic records reviewed met the criteria for inclusion in this research." (lines 322-323).
Lines 311-312. However, their results contradict this statement because the results of this Coopers Hawk study demonstrate that arthropods are well recorded, even with frequencies strongly higher than vertebrates.
Lines 324-325. Ok, but can provide some examples of these different approaches used for the study of rare or cryptic species.
Conclusions
I consider that the conclusions are not articulated with the stated objectives of the study because one of the two stated objectives was not analyzed, and was only presented visually through Fig. 1, but lacking any description and analysis.

Reviewer 2 ·

Basic reporting

Largely well-written, English language corrects needed in some areas of the text but I did not struggle to understand what was being conveyed/present. Literature is generally good, however, there are a few important and very relevant studies missing from the reference list. I have included this in my review for the author’s to consider. Overall, the figures and tables are good, however, please see my detailed comments with recommendations on improving the communication of results. Raw data is shared. The biggest weakness of this manuscript, in my opinion, is the lack of links between the results and any testable hypotheses. I have commented on this below, specifically with regards to the end of the introduction.

Experimental design

The original primary research is indeed within the aims and scope of PeerJ, pending revision I believe this study would be of interest to the journal’s readership. Research questions missing and currently this aspect of the study is inadequate. Explanations on how this research fills identified knowledge gaps requires improvement. Given the small sample sizes, I understand why no formal statistical analyses, e.g., linear modelling, were conducted. However, there was scope for the authors to use more descriptive statistics to explain variation within their data sets, e.g., ANOVAs. Methods section requires improvement, see below.

Validity of the findings

More clarification needed in the methods before this study can be considered completely replicable. Underlying data has been provided and there were no obvious inconsistencies present within it. Discussion and conclusions section need revising. Currently, there are far too many results presented in the discussion for the first time and not enough discussion around the strengths and weaknesses of the method, and what the authors’ results mean for our understanding the ecology of these species, with explicit links to relevant published research.

Additional comments

Title is a little vague. A strength in your study is the use of your methods, therefore, I suggest changing it to “Diet composition of four raptor species in Mexico using a combination of web-sourced photography and literature searches”.
58 – reference needed for this statement.
65 – ecological niches?
70-72 – provide species common names before scientific names.
85-88 – context around why you chose these species in particular is missing here. Surely, there are many raptor species that are under-represented in research from Mexico, so why these specific species? Was it because they represented the most under-studied species? Had adequate volumes of photographic data online? Clarity needed here.
93 – okay the introduction is lacking relevant information for the reader here. I think you should include a paragraph of two on different methods of studying raptor diets and why web-sourced photography is the best method to use in this study. Highlight its strengths but also do not ignore the limitations to ensure the reader fully understands why you use this method and for the species included. The end of the introduction is currently very weak, what were your specific research questions/hypotheses? What do you expect to find in relation to previously published research? What is the rationale for doing this study? I don’t think encouraging people to engage with citizen science platforms is the best rationale for this study. The general public are unlikely to read this article. Your target audience is other scientists/ornithologists. So perhaps, like you say, your findings fill outstanding knowledge gaps on the dietary ecology of the species studied here… with more traditional methods, e.g., analyses of pellet/prey remains, being ineffective to fill these gaps. Hence why you used web-sourced photography… expand into benefits of it as a method.
92 – again, the geographic coverage rationale is missing. Why did you look at spatial spread of dietary information? Did you do any analysis on this, etc? What is the benefit to the knowledge on the ecology of these species that analysing dietary data spatially provides? This needs to be included here.
100 – provide the (%) too.
101 – scientific names have already been defined in the introduction.
103 – wide distribution does not always necessarily equate to more photographs. It depends on the species’ ecology, habitat preferences and morphology. E.g., larger birds tend to be more photographed (https://www.nature.com/articles/s41598-021-98584-7.pdf) and biases towards those from urban or more human environments (https://onlinelibrary.wiley.com/doi/full/10.1111/ibi.12918)
111 – (Davis, 2014)
129 – reference needed.
130 – grey or gray? Choose one and be consistent.
142 – “richness of prey” I am conflicted as to whether this is a good phrase to use. I automatically assume you mean species richness, i.e., simply number of different species, in their diets? Perhaps simply “diets” is better throughout as you are interested not only in number of prey species but also dietary composition.
144-145 – why not Macaulay Library instead of eBird?
153 – “In order to avoid record overestimation, we only considered one predation record per day, per location, per raptor species, per raptor age (the studied species have an age-related polymorphic plumage), per prey.” I do not understand the rationale for this at all. How do you define “location”, excluding all these records surely will impact your ability to provide a comprehensive overview of the species’ diets? Checking for duplicates by author name/date/location would be more than sufficient… This is also the first-time raptor age classifications have been mentioned. This warrants its own section. I am also surprised that you decided to exclude photographs where the prey could not be identified, these unknowns are still important in accurately describing the species’ diet as they provide an estimate of how effective your methodology was at describing prey.
161 – why did you not use Web of Science which is the go-to database for literature searches?
173 - you extracted month and year data but do not mention any temporal aspect to your analyses at the end of the introduction.
185-186 – again, not mentioned at the end of the introduction where I would expect to find the specific research questions.
190 – which metric was specifically used? Levin’s niche breadth (Bn)? Bn[j] = (1/R)/sum(p^2) whereby R is the number of differing environments and p is the proportion of taxon j in environment i.
195-205 – providing percentage values alongside the raw numeric values in text would really help orientate the reader here.
206-210 – these results are very descriptive and I would strongly recommend trying to present the importance of each main prey group in the diet of each raptor species. E.g., presenting these findings as relative contributions in the diet of each would make more sense to me. Currently, there are no numbers at all in your results that support statements here.
207 – it did not change, simply differed between species.
210-212 – provide the niche breadth values here
218 – were range maps sourced from BirdLife datazone? I do not recall this being specifically mentioned in the methods. Anyway, I’m not sure if this finding is that relevant, as the polygon ranges are rather coarse and not really suitable for very fine-scale analyses.
225 – results are presented in the discussion, these need to be moved. The data presented here are missing in the results section!
242 – again, all of this needs to be in the results section not here.
271-273 – I’m not sure Roadside Hawks and White-tailed Eagles are directly comparable? Can you compare your findings for this species with another Accipiter hawk from the Americas instead?
281-285 – expand on the ecology of these two species which may reflect the observed patterns in your dietary data.
300 - I guess the elephant in the room here is whether the species is abundant enough in this part of its range? Species abundance is not uniform throughout its geographic range, and thus there are often large areas of suitable habitat within the range where the species does not occur or occurs in low densities, which may explain lack of data. Also, bear in mind observer bias and accessibility to these areas. Did you test for a urban land cover effect in the spatial distribution of your photos? It would be interesting to see whether the distribution of your photographs is correlated with human footprint index data, i.e., assume a positive relationship between number of observations and grid cells with higher HFI scores. This would be a straight-forward analysis to do.
324 – such as what? DNA metabarcoding, remote cameras on nests, etc?
334 – this may simply be an artifact of your sampling strategy, using Google Images, Social Media photos and other repositories (WikiAves [if not restricted to just Brazil], etc) will likely increase spatial coverage of your data set.
Some raptor dietary studies that use web-sourced photographs have not been cited and thus are missing from the reference list, consider:

o Berryman & Kirwan (2021) Journal of Raptor Research https://doi.org/10.3356/0892-1016-55.2.276
o Panter & Amar (2021) Ibis https://doi.org/10.1111/ibi.12918
o Kannan et al. (2023) Acta Ornithologica https://doi.org/10.3161/00016454AO2022.57.2.004
o Pantoja-Maggi et al. (2024) Journal of Raptor Research https://doi.org/10.3356/JRR-23-32
Figure 1 –consider changing the y-axis title to “Proportion of diet (%)”.
Figure 2 – nice maps, but reconsider colour gradient. Red-green can be very difficult to see for colourblind people like myself. A blue gradient may suffice, with the species’ distributions in a lightish grey?
SI – you collected temporal data but did not do any analysis on this? I would expect a figure showing the distribution of your samples over time (years and months).

Reviewer 3 ·

Basic reporting

The authors clearly define the importance of diet analysis to understand the biology of the species and their role in the ecosystem. They also highlight important aspects of raptor diet: their role as predators, interactions with other species, conservation risks, their relevance in rehabilitation and how knowledge of diet can help in management and conservation. However, they could enrich the article by emphasizing the global importance of studying the diet of raptors, or how changes in diet can be an indicator of broader environmental changes (e.g., climate change, habitat loss), across the guild distribution of the species studied. As well as briefly highlighting the importance of raptors in the ecosystem (i.e. population control, seed dispersal, etc.). It is suggested to move the information on the family Accipitridae (currently in the second paragraph) to the first paragraph, to help establish the territory of the topic.
The structure of the introduction can be strengthened by relating conservation implications to the lack of knowledge. Instead of just mentioning that “information is lacking”, it is suggested to argue that this makes it difficult to implement effective conservation strategies, to assess their vulnerability to environmental changes or to develop management plans that ensure their long-term survival. It is also necessary to incorporate the research question posed in the research. It is also necessary to consider adding the hypothesis proposed, based on previous studies, in order to have a clear idea of the expected results and to clarify the proposed objective. Consequently, it is necessary to explain the relationship of the analysis of the geographical coverage based on the records with the research question that governs the study.

Experimental design

The research presented is original and innovative. Although the authors explain how the research fills an identified knowledge gap, the research question that drives the study and how it relates to both the determination of diet with non-traditional methods and the analysis of the geographic coverage of prey records is not clearly defined. Therefore, the research question and hypothesis driving the study should be added and how the study contributes to addressing it should be explained.
The research makes a valuable review on the subject, which is the basis for the cleaning of the main data. However, the method described needs to be more detailed to ensure replication. It is necessary to give details in the “Geographical coverage of data” section and in the “Statistical analysis” section. For example, give details of the type of niche breadth index used and describe it briefly. Likewise, detail how the prey categories and frequencies were taken for the analysis, as well as explain whether the items considered in the index calculation were grouped. In the case of the “Study Species” section, I consider that the description of each species should be contextualized, highlighting the general characteristics used for the identification of the species, indicating the guides used as references, etc. Likewise, it is recommended to detail the distribution of the species for Mexico, in addition to the reference to the general distribution that is made.

Validity of the findings

The conclusions of the study reflect the results presented, although it would be valuable to strengthen their explicit connection to the original research question. We acknowledge the considerable effort devoted to the collection and cleaning of dietary records. However, the inherent limitations of data derived from citizen science platforms pose significant challenges to the robustness of the conclusions, particularly when interpreting trophic niche breadth at a regional scale or for the entire distribution of the species in Mexico.
Given the small number of predation records per species, interpretation of trophic niche breadth should be approached with caution. Patterns of specialization or generalization in diet may not be robust and could vary substantially with increasing completeness of samples or records, especially considering the vast range of these raptors. Trophic niche breadth is a key ecological concept for understanding the species-environment relationship and predicting the response of species to environmental changes, so its correct interpretation has direct implications for conservation, distribution and colonization capacity. The authors could substantiate the validity or scope of the amplitude index at this scale.
In this sense, I suggest evaluating the possibility of focusing the study on the description of diet composition. A more descriptive approach would allow highlighting the identification of prey not previously reported in studies conducted with traditional methods. To maximize the impact of the discussion, it is suggested to focus on the detailed description of diet composition, relating these findings to the natural history of the species and ecosystem characteristics. Interpretation of trophic niche breadth could be reserved for future research, with more robust data sets. Emphasizing the added value of records obtained through citizen science, specifically in expanding the inventory of prey consumed by these raptor species, could significantly strengthen the contribution of the study.
Finally, while the authors suggest the feasibility of this methodology for frequent and widely distributed species, the low proportion of cells in the potential distribution that have records raises questions. It would be valuable for the authors to further discuss the conditions under which this approach, based on citizen science, might be more successful with other raptor species, perhaps exploring the data threshold needed to obtain robust results.
The structure of paragraphing should be carefully reviewed, especially in the discussion section, as well as the format of in-text citations.

---

## Round 0.2 · Major Revisions

Dear Authors

Thank you for submitting the revised version of your manuscript entitled "Urban areas promotes shifts in the proportion of prey consumed by four raptor species (Accipitridae) in Mexico" to PeerJ.

After evaluating the updated manuscript and carefully considering the new round of reviewer feedback, I regret to inform you that major revisions are still necessary before your submission can be considered for publication.

In particular, Reviewer 2 noted that several of their key concerns raised during the first round of review were not addressed in the revised version. The reviewer 2 emphasized that many of their substantial comments were either overlooked or insufficiently responded to.

Given the nature of these unresolved issues, I strongly encourage you to either revise the manuscript to comprehensively address all reviewer concerns or provide a clear and detailed rationale explaining why certain changes were not implemented.

We appreciate your interest in publishing with PeerJ and look forward to receiving your thoroughly revised manuscript along with a detailed response to all reviewer comments.

Best regards,
Armando Sunny

Reviewer 1 ·

Basic reporting

No comment

Experimental design

No comment

Validity of the findings

No comment

Additional comments

The authors responded to each of the comments clearly and concisely.
Therefore, I have no problem accepting this latest version of the manuscript.

Reviewer 2 ·

Basic reporting

Please see additional comments.

Experimental design

Please see additional comments.

Validity of the findings

Please see additional comments.

Additional comments

I am not wholly convinced that the authors have considered all of my original concerns with the manuscript. It appears that the authors may have cherry picked the revisions that they deemed easier to do and passed on the more extensive revision requests. I will reiterate my major concerns with this study again:

1) Nowhere in the manuscript do the authors control for or at least demonstrate to the reader that their samples may be biased towards urban areas. In the first round of comments, I asked the authors to seriously consider exploring the relationship between the spatial distribution of their photographic samples and some form of urbanisation metric, e.g., human footprint index, which is easily accessible online. In their response, I could not help but notice that this suggestion was ignored. The scope of the study has shifted towards urban vs. non-urban diets but they do not provide any quantitative data to show whether their samples are biased.

2) I do not think a GLM is appropriate here. Given their total sample size across all species is 400 photographs, I remain unconvinced that they have sufficient samples within their categorical data levels to compute a robust, statistically sound analysis here.

3) Current descriptions of the species’ diets are heavily biased and dependent on sampling effort employed by the authors. Specifically relating to the filtering of photographic records, which I do not understand. You cannot provide an overview of a species’ diet based on a pre-selected sample. I assume this was done to prevent spatial autocorrelation, however, I suggest that the authors apply a spatial thinning approach on a full dataset before any spatial analyses but retain the full dataset when describing diet(s).

Other comments:
The focus on “studying diets for conservation” is inappropriate here, as all four study species are globally least concern and there are no conservation-related data presented. I previously asked the authors to reconsider this.

Literature searches – no information provided on title, abstract and full article screening. No evidence of refining search strings and how this differed between repositories. Google Scholar will provide thousands of returned results, how did the authors decide when “enough was enough”? Were only peer-reviewed articles included in the eligibility criteria? What about grey literature sources, review articles and other article types (assume the two latter were excluded)?

Minor comments (tracked changes version):
Title – I’m unsure that the title is accurate given the inherent biases in the dataset mentioned throughout.
L16 – again, I am still unsure about this. Perhaps diet information is relevant for species with declining populations or those of conservation concern, but how does the conservation aspect link to your selected species here? All of your study species are globally Least Concern. The conservation aspect doesn’t work for me.
L28 – “diets” of these species. Frequency of prey is too refined here.
L42 – it didn’t “remain constant”. Simply, it did not differ statistically. Rewording needed here.
L46 – in which direction? This needs specifying here.
L49 – any proportional values to back this up here? E.g., from an average across species of XX% to XX% in urban and non-urban areas, respectively.
L54 – revised abstract stops rather abruptly and short of a concluding sentence focused on the key take home message of this study.
L64 – I strongly advise including “web-sourced photography” as a keyword.
L67 – I suggest leading with this opening point/sentence in your abstract.
L76 – new paragraph.
L78 – see comment above regarding conservation.
L80 – pellets, prey remains and direct observations. Cite some studies here as a couple of examples.
L83 – see papers by Ryan Bourbour et al. for some nice examples of this in action (and outside of the breeding season).
L91 – awkward link from web-sourced photography as a tool to study raptor diets and the taxonomy of Accipitridae… please reconsider the structure and flow here.
L102 – it is unusual to cite a study that is nearly 20 years old to describe research effort to date. Can you find some more recent examples for these species?
L109 – another awkward link, please improve the flow from research in Mexico and urbanisation effects on raptors. Perhaps some restructuring of the entire introduction is needed. This should go further up, after you introduce raptors as a group.
L150 – “prey species richness”.
L155 – only for individuals that are successful in rearing young, this does not include floaters, failed breeders or immature individuals that are yet to come into sexual maturity.
L158 – typo “hereafter”.
L163-166 – this should be first mentioned towards the end of the introduction. Line 163 to the end of this paragraph should be moved. Provide a broad description of habitat types in the study area in replacement of this section.
L174 – reference needed.
L176 – I think you should only describe the distribution for each species in Mexico, as you have already filtered down to your study area in the previous paragraph.
L182 – breeding season in Mexico, right?
L223 – typo/repetition.
L225 – provide a clean URL for Macaulay Library, i.e., https://www.macaulaylibrary.org/
L234-237 – I’m not quite sure about this, what was the reason for this? “To avoid record overestimation” is vague. Is this to do with spatial autocorrelation? I have concerns that due to your filtering, you are not capturing a fair assessment of the diet for each species. You could thin the data spatially before any spatial analyses to avoid autocorrelation, but for the dietary overview I would expect the data to come from a full set of observations, not filtered.
L239 – which record? The most recent? Best quality image? How can someone repeat this method using the information provided?
L257 – in Web of Science, did you
L264 – “plotted” instead of “projected”. Projected would imply you reprojected the spatial point data into an appropriate coordinate projection. Actually, that raises an important point, which CRS and projection were used for this?
L268 – methods are too vague here. Which land use classes were used, did you have to reclassify these? What is actually included under “urban areas”?
L281 – version needed, I assume 4.4.1?
L285-290 – I am still unsure what “sample coverage” refers to here, even after reading this section. More details needed please, perhaps a definition from Chao & Jost (2012)?
L295 – I am confused again. So you plotted the (filtered)data points in a GIS, plotted the vector polygon range maps for each species and then what? How was geographic coverage defined? Did you create a new polygon using all data points for each species and calculate percentage overlap between this and range maps?
L310 – In my previous comments, I suggested that the authors use more descriptive statistical tests to analyse patterns in the data due to the limited sample sizes. It seems that the authors have employed linear modelling as an approach here, however, I have concerns that their data may violate the assumptions of this technique. My main concern is around having adequate samples for each species and covariate combination, especially with the data filtering applied beforehand. It would help to have a summary table of all sample sizes (including %) for each species and covariate. Are the authors sure they had enough (>30 observations) for each species outside of the breeding season to reliably test the effects of season on their response variable? Again, are there sufficient samples for each species and each month to test this? The text in the methods explaining the modelling again is lacking sufficient detail. What do the authors mean by “reference variables” on L314?
L331 – my point exactly. Your entire dataset contains 401 observations for four species. I am unconvinced that you have sufficient sample sizes for linear modelling here.
L341 – I am still unsure what sample coverage means here.
L348 – right now I understand what geographic coverage means, however, this is not adequately described in the methods.
L366 – no description in the methods on how the authors classified prey species to taxonomic group. Where are the “unknowns” and why are they not included? I explained why including unknown prey species was important in my previous round of comments, this appears to have been ignored. Your dietary data are biased because of a) the filtering applied, and b) exclusion of unknowns and thus only represent a picture of the species’ diet from a limited pool of photographs where you could only identify the prey species. Similarly, I would be interested in knowing what the samples sizes are for each prey group for each species, insert this in the text when presenting results: e.g., mammals (n = XX) P < 0.0001 vs. reptiles (n = XX) P < 0.0001. What do these P values refer to here? Surely if you are doing post-hoc comparisons then you only have a single P value for each comparison, e.g., mammals vs. reptiles P < 0.0001. We also need the test statistic included here to know the direction of the relationship. It is always better to include the actual P value instead of abbreviating to < 0.0001.
L389 - revise to (Mann-Whitey: U = 15, df = XX, P < 0.001). Include degrees of freedom when presenting all statistical results in text. Degrees of freedom can be calculated from your sample size minus 1.
L430 – see previous comments, re conservation.
L432 – what do the authors mean by “studying raptor diets has implications for the ecological dynamics of their prey” some clarification needed here.
L433 – not so sure about that? There are many papers that explore the diets of urban raptors, especially in recent years. Like I said before, the unique point of this study is that it uses a relatively new method and this should be the main emphasis.
L438 – results in discussion.
L457 – this is the first time the authors have mentioned this. I would expect this to be in the methods and results section, not here for the first time. I think you need to include the numbers and percentages for each prey group where you were able to identify prey to species level.
L463 – because the authors filtered out most of the records(!).
L498 – singular “prey”.
L508 – again first time these results are presented in text, in the discussion?
L516 – species names needed. At least genera-level if multiple species.
L520 – Panter & Amar (2021) did not conclude that raptors prey on exotic species. The top prey species in their study were all native to their study region. Keep Cava et al. (2012).
L528 – how common is hanging bird feeders in Mexico? Is this a regular practice like elsewhere in the world, e.g., Europe?
L574 – you can cite Newton here.
L589 – I strongly think that in the case of this study, which describes diet, all photographs with prey should have been processed with no filtering.
L688 – crustaceans? Such as which species?
L703 – if you hadn’t had filtered then your sample sizes would have been much larger and more representative of the species’ diets.
L708 – cite Bourbour et al’s work here, and a study that has used stomach contents analysis. However, the latter wouldn’t be suitable for dietary studies that extend across large spatial or temporal scales as the bird needs to be dead before examination.
Figure 1 – really nice maps.
Figure 5 – needs a legend on the plot showing what each shading/pattern represents.
Figure 6 – same here.

---

## Round 0.3 · Major Revisions

Dear Authors,

Thank you for submitting the revised version of your manuscript entitled “Urban areas promotes shifts in the proportion of prey consumed by four raptor species (Accipitridae) in Mexico”. We appreciate the effort you have made in addressing the initial round of comments. However, after careful evaluation of the revised manuscript and consideration of the reviewer’s feedback, we find that several key issues remain insufficiently resolved.

In its present form, the manuscript still raises concerns regarding the robustness of the methodology and the reliability of the conclusions. Therefore, we must request major revisions before the work can be considered further. The reviewer’s main points are summarized below, and we ask you to address them thoroughly in your next revision:

Statistical modelling and sample sizes
While your dataset includes approximately 400 records, some species × habitat combinations remain very sparsely represented (e.g., Buteo plagiatus in urban areas, n = 18; Buteogallus anthracinus in urban areas, n = 23). Estimates derived from such small samples are unlikely to be stable or representative when extrapolated to the national scale. We strongly recommend that you:

Restrict analyses to categories with sufficient sample size,

Collapse categories with sparse data, or

Explicitly acknowledge and discuss the limitations of categories with <30 records.

Urban bias in photographic data
The concern regarding spatial sampling bias remains unaddressed. Web-sourced photographs are inherently more likely to be collected near people (urban, peri-urban, roadsides, or easily accessible protected areas), and thus the sampling distribution is not random. Without a quantitative assessment of sampling bias (e.g., comparing record locations to human footprint, road density, or settlement proximity), it is not possible to know whether the observed patterns reflect raptor ecology or observer effort. A clear methodological assessment of this potential bias is required.

Exclusion of unknown prey items
Excluding unidentified prey introduces a potential bias in prey representation, as groups that are harder to identify (e.g., small reptiles, amphibians, invertebrates) are systematically omitted. Even if “unknown” is not a biological category, reporting the proportion of unidentified prey is important for methodological transparency. Please quantify and report these cases, and discuss their implications for the representativeness of the dataset.

Filtering steps and transparency
Earlier versions of the manuscript mentioned filtering steps to avoid record overestimation (e.g., one predation record per date, location, species, prey). This methodological step seems to have been removed without explanation. Please clarify whether this filter was applied in the final analyses and justify your decision.

Taken together, these points highlight important methodological and interpretative issues that must be resolved to ensure the study’s conclusions are statistically and ecologically sound.

We encourage you to carefully address each of the concerns above in a substantially revised manuscript, providing additional analyses, clarifications, and transparent discussion where appropriate.

We look forward to receiving your thoroughly revised version.

Sincerely,
Armando Sunny

Reviewer 2 ·

Basic reporting

See additional comments.

Experimental design

See additional comments.

Validity of the findings

See additional comments.

Additional comments

Statistical modelling
Thank you for the clarification and for adding model fit statistics. I remain concerned, however, that the robustness of your GLM analyses is undermined by the distribution of sample sizes across categorical factor levels. While the overall dataset comprises ~400 observations, several species × habitat groups have relatively few records (e.g. Buteo plagiatus in urban areas, n = 18; Buteogallus anthracinus in urban areas, n = 23). GLMs can technically be fit with small samples, but estimates based on such limited information are unlikely to be stable or biologically representative, particularly when extrapolated to the scale of the entire country.

In other words, although the total dataset appears sufficient, the effective sample sizes for some categories are too small to support robust inference about urban vs. non-urban diets across Mexico. AIC and R² values describe how well a model fits the observed data, but they cannot compensate for the lack of information in underrepresented categories.

I strongly recommend that the authors either: 1) restrict their modelling to species and habitat groups with adequate sample sizes, 2) collapse sparse categories to ensure sufficient replication, and 3) clearly acknowledge in the discussion that results for species–habitat combinations with <30 records should be interpreted with extreme caution and cannot be considered representative at national scale.
Without this, there is a risk that readers will take the results at face value, overestimating the reliability of patterns driven by very small sample sizes.

Urban bias
Thank you for the detailed reply. I appreciate the effort to map records against urban polygons and test for urban vs. non-urban differences. However, my original concern was not about differences in the number of predation events among species, but rather about bias in the underlying data source.
Web-sourced photographic data are inherently biased toward areas where people live and take photographs (typically urban and peri-urban areas, roadsides, or protected areas with easy access). This means the probability of an observation being recorded is not equal across space. As a result, your dataset is more likely to over-represent feeding events near human settlements, regardless of the actual ecology of the raptors.

The fact that you found more non-urban records for some species does not remove this bias, because those numbers still depend on where people happen to photograph raptors. What is missing is any demonstration that the spatial distribution of sampling effort itself is not skewed toward urban environments. A simple way to address this would be to compare the spatial distribution of your records against an independent measure of human influence (e.g. human footprint, road density, or distance to settlements). Even if you do not use the human footprint index per se, some quantitative assessment of sampling effort bias is essential.

Without this, readers cannot separate ecological patterns from methodological artefacts. Given that your study aims to evaluate raptor diet “in urban vs. non-urban areas across Mexico,” but some categories are represented by very small sample sizes (e.g. only 18 urban records for Buteo plagiatus), I am concerned that the results risk being presented as ecological patterns when they may partly reflect uneven observer effort. At a minimum, this limitation needs to be explicitly acknowledged in the discussion, and ideally, a spatial analysis of potential sampling bias should be attempted.

Unknown prey items
Thank you for your detailed reply. However, my concern has not been fully addressed. I am not suggesting that “unknown” prey items should be treated as a biological category equivalent to mammals, reptiles, etc. Rather, the point is that the exclusion of unidentified prey items introduces a prey identification bias into your dataset and subsequent models, because the analysis is then restricted only to photographs in which prey could be confidently identified. This biases your results towards certain prey groups that are more readily identifiable (e.g., mammals with hair, birds with feathers) and excludes others that may be under-represented simply because they are harder to identify in photographs (e.g., small reptiles, amphibians, invertebrates).

The inclusion of “unknown prey” as a category is not about biological interpretation but about transparency and methodological accuracy. Unknowns provide an estimate of the effectiveness (and limitations) of your methodology. Without them, the reader cannot evaluate the representativeness of your results. For example, if 40% of prey records are excluded as “unknown,” then the dietary composition presented only reflects a subset of the species’ diet and cannot be considered a full picture. This is a critical point because it affects the strength of the inferences you are drawing.
I therefore strongly encourage you to explicitly acknowledge this limitation in the Methods and Discussion, and either (i) include unknown prey as a reported category (even if excluded from statistical tests), or (ii) clearly quantify and justify the proportion of records excluded due to being unidentifiable, and discuss how this may bias your conclusions.

Other points
Filtering of photographic data – in earlier versions of this manuscript, the authors stated that “In order to avoid record overestimation, we only considered one predation record per date, per location (location, name and coordinates), per raptor species, per prey.” This is the filtering that I was referring to. It appears to me that this has now been removed from the manuscript without explanation.

From response letter: “Our response to your first major concern provides a detailed explanation of how we prove that our dataset is not biased toward urban environments”. The authors have not proven that their data is unbiased towards urban environments whatsoever, I suggested a number of ways they could demonstrate this which again have been ignored/dismissed.

---

## Round 0.4 · accepted · Accept

Dear Dr. Márquez-Luna,

Thank you for your submission to PeerJ.

I am writing to inform you that your manuscript - Urban areas promotes shifts in the proportion of prey consumed by four raptor species (Accipitridae) in Mexico - has been Accepted for publication. Congratulations!

Armando Sunny

Reviewer 2 ·

Basic reporting

Please see additional comments.

Experimental design

Please see additional comments.

Validity of the findings

Please see additional comments.

Additional comments

Thank you to the authors for carefully considering my previous round of comments. I am satisfied that the majority of my previous concerns have been dealt with, and the additional test for potential urbanisation bias adds a layer of clarity to the study which I felt was lacking before. My only remaining concern relates to the sample sizes used in the linear modelling. If it were me, I would not proceed with the categories and samples presented. However, to avoid going around in circles and for the interests of both parties, I suggest we leave this issue to the editor's discretion. Good luck with the remaining process and all the best.